# Relationship between stair ascent gait speed, bone density and gait characteristics of postmenopausal women

**Ali Dostan** [1]* *, **Catherine A. Dobson** [1] *, **Natalie Vanicek** [2] *

**1** Biomedical Engineering Research Group, School of Engineering, University of Hull, Hull, United Kingdom,
**2** School of Sport, Exercise and Rehabilitation Sciences, University of Hull, Hull, United Kingdom

* These authors contributed equally to this work.
* A.Dostan@hull.ac.uk

**Data Availability Statement:** Data are all contained within the manuscript and the supporting information file.

**Funding:** This research was funded through a charitable donation by Osteoporosis Research in

## Abstract

Stair ascent is a biomechanically challenging task for older women. Bone health may affect gait stability during stair walking. This study investigated the gait biomechanics associated with stair ascent in a group of postmenopausal women in relation to walking speed and bone health, quantified by T-score. Forty-five healthy women (mean (SD) age: 67 (14) years), with bone density ranging from healthy to osteoporotic (T-score range +1 to -3), ascended a custom-made five-step staircase with two embedded force plates, surrounded by 10 motion capture cameras, at their self-selected speed. Multivariate regression analyses investigated the explained variance in gait parameters in relation to stair ascent speed and T-score of each individual. Stair ascent speed was 0.65 (0.1) m·s$^{-1}$ and explained the variance ($R^2$ = 9 to 47%, $P \leq 0.05$) in most gait parameters. T-score explained additional variance in stride width ($R^2$ = 20%, $P$ = 0.014), pelvic hike ($R^2$ = 19%, $P$ = 0.011), pelvic drop ($R^2$ = 21%, $P$ = 0.007) and hip adduction ($R^2$ = 7%, $P$ = 0.054). Increased stride width, and thereby a wider base of support, accompanied by increased frontal plane hip kinematics, could be important strategies to improve dynamic stability during stair ascent among this group of women. These findings suggest that targeted exercises of the hip abductors and adductors, including core trunk musculature, could improve dynamic stability during more challenging locomotor tasks. Balance exercises that challenge base of support could also benefit older women with low bone mineral density who may be at risk of falls.

## Introduction

Negotiating stair ascent is more physically and mechanically challenging than level walking [1] and presents a greater risk of falling, especially for older women [2, 3]. A fall on stairs can have serious consequences, and in the most severe cases may lead to hospitalisation and loss of independence and quality of life [4]. Osteoporosis increases the likelihood of fractures at the hip, wrist and vertebrae [5]. Although changes in bone mineral density (BMD) cannot be considered the sole risk factor in osteoporotic fractures [6], a previous study identified that loss of

East Yorkshire (OSPREY) (charity commission number: 1013289). The funders had no role in study design, data collection and analysis, decision to publish, or preparation of the manuscript.

**Competing interests:** The authors have declared that no competing interests exist.

BMD in the femoral neck region was an important predictor of fracture risk in older women [7]. Day-to-day load-bearing activities, such as stair climbing, can help to maintain and attenuate further bone loss in older adults [8].

Negotiating stairs safely becomes more challenging as we age. The underlying biomechanical reasons are associated with decreased musculoskeletal capacity and consequently reduced gait speed [9]. Older adults have demonstrated greater hip frontal plane moments when compared to younger adults and maintain lateral stability by relying mainly on the hip abductors [10]. They also operate at a higher relative capacity compared to younger adults when it comes to utilising extensor moments [9] and consequently develop compensatory strategies to meet the biomechanical demands of stair ascent. For example, by redirecting energy from the knee distally towards the ankle, they are able to generate a greater plantarflexor moment [9]. It is unclear whether different compensatory stair ascent strategies exist for older women with low BMD, as a fall could have more severe consequences in this group (e.g., a fracture).

There are a limited number of studies that have investigated the relationships between level walking gait parameters and BMD in older women [11–16]. To the best of our knowledge, no study to date has investigated the biomechanics of stair ascent in relation to low BMD or osteoporosis. Research in this area would help make evidence-based recommendations for improved functional capacity of older women with low BMD during common daily activities. The aim of this study was to explore the gait, biomechanics in relation to comfortable stair ascent speed and T-score (a standardised level of BMD) in older women with a broad range of T-scores, spanning from healthy to osteoporotic, during stair climbing. It was hypothesised that speed would be the most important predictor and explain most of the variance in both joint kinematic and kinetic parameters. However, it was anticipated that inclusion of the T-score in the regression model would explain additional variance for kinematic and kinetic parameters, especially related to the knee and ankle in the sagittal plane, and pelvis and hip in the frontal plane.

## Methods

### Participants

Forty-five healthy postmenopausal women, aged between 65–70 years with a BMI between 18–30 kg/m$^2$ and various levels of BMD (ranging from +1 to −3 T-score), were recruited from the local Centre for Metabolic Bone Disease. T-score compares the BMD at a specific site with that of a young, healthy sex-matched group and expresses the relative level of BMD as a deviation from the mean peak value. The same technician measured the BMD (expressed as T-score) at the femoral neck with a DXA scan for all participants, resulting in n = 13 with healthy BMD (-1 SD ≤ T-score ≤ +1 SD), n = 26 with osteopenia (-2.5 SD ≤ T-score ≤ −1 SD) and n = 6 with osteoporosis (T-score ≤ −2.5 SD). Therefore, 29% of the participants a healthy BMD, while 71% were considered to have low BMD.

Eligibility criteria were set to exclude any participant who presented with gait abnormalities, neurological disorders, any cardiac failure, or if they had received a treatment course of hormone replacement therapy, glucocorticoids, teriparatide and/or bisphosphonate within the five years prior to study enrolment. Favourable ethical opinion was granted by the NHS Local Research Ethics Committee (Ref. 11/YH/0347) and all participants gave their written informed consent prior to participation. Participants' demographics are reported in Table 1. Based on the World Health Organization guidelines (WHO) [17], 81% of participants self-reported achieving at least 150 minutes of moderate intensity, or 75 minutes of vigorous intensity, or a combination of both, of physical activity every week.

**Table 1. Participant (n = 45) characteristics.**

|  | Mean (SD) | Range |
|---|---|---|
| Age (years) | 67.3 (1.4) | 65 to 70 |
| Height (cm) | 161.4 (4.9) | 151 to 172.5 |
| Mass (kg) | 63.5 (8.6) | 47.8 to 80.4 |
| BMI (kg/m$^2$) | 24.1 (2.8) | 18.6 to 29.2 |
| Femoral neck T-score | -1.5 (0.8) | 1 to -3 |
| Number of days physically active [a] (days per week) | 5 (2.3) | 0 to 7 |
| Commencement of menopause (age in years) | 50 (4.5) | 38 to 58 |
| Number of falls (last 12 months) | 1 (0.6) | 0 to 3 |
| Number of fractures (>50 years old) | 1 (0.9) | 0 to 4 |

[a] Activities included walking, Zumba, badminton, golf and general gym exercises.

## Protocol

Participants wore their own tight-fitting clothing and normal, flat walking shoes during one visit to the laboratory. Forty-four retroreflective markers (14 mm), including clusters of four markers on the thigh and leg, were secured bilaterally onto the lower limb segments according to the six degrees of freedom (6DoF) marker set [18].

Three-dimensional kinematic data were collected using twelve Pro-Reflex MCU1000 motion capture cameras (Qualisys, Gothenburg, Sweden) sampling at 100 Hz while participants ascended a 5-step custom-built wooden staircase (step height: 20cm tread 30cm, width: 80cm) surrounded by a separate wooden handrail structure (Fig 2). Two Kistler piezoelectric force plates (model 9286AA, Kistler, Winterthur, Switzerland) were embedded into the second and third steps of the stairway and synchronised with the motion capture system. Ground reaction forces (GRFs) were sampled at 500 HZ. Participants walked along a 5-metre level walkway to achieve a steady pace before ascending the staircase at their preferred pace and 10 trials were recorded. No adverse events (trips or falls) occurred.

## Data analysis

Marker coordinates were first processed in Qualisys Track Manager Software (version 2.09) before being analysed in Visual 3D v3.0™ (C-Motion, Rockville, USA). 3D marker coordinate data were interpolated using a cubic-spline algorithm. Marker trajectory and GRF data were filtered using a low-pass 4$^{th}$ order Butterworth filter with a cut-off frequency of 6 Hz and 25 Hz, respectively. A lower limb, 7-segment 6DOF model was built in Visual 3D based on the static calibration file with bilateral virtual feet segments and CODA and Visual 3D pelvis segments. Gait speed was computed in Visual 3D using the actual stride length / actual stride time. Joint moments and powers were calculated using inverse dynamics analysis. The ankle joint moment was calculated considering the effects of the gravitational force on the centre of mass, and GRF acting through the centre of pressure as well as the joint reaction force [19]. The segment's centre of mass location and moment of inertia were based on Dempster's (1955) values [20]. The results were then incorporated to determine the subsequent proximal joint moments. Joint moment and angular velocity were utilised to determine the joint powers. The X-Y-Z Cardan sequence defined the order of rotations following the Right Hand Rule about the segment coordinate system axes.

Stair ascent gait data were normalised to the gait cycle. The trail limb data started with foot contact on the second step and terminated on the fourth step. The lead limb data started with

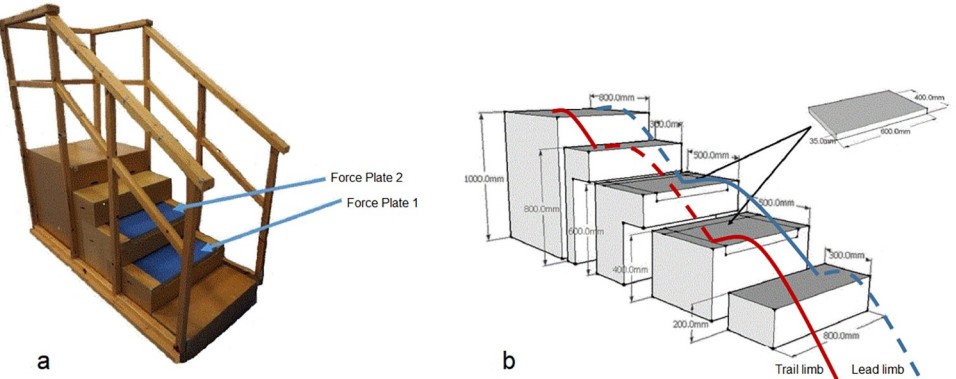

**Fig 1.** a) Laboratory staircase, force plates 1 and 2 were embedded in the second and third steps of the staircase, respectively, b) Schematic demonstration of the lead (blue line) and trail (red line) limb gait cycles during stair ascent.

foot contact on the third step and terminated on the top, fifth step. This allowed us to analyse steady-state stair ascent for both limbs. Gait events (initial contact and toe-off) of the lead and trail limbs were identified using the kinetic data from the force plates embedded into the third and second step, respectively (Fig 1). The foot contact terminating the gait cycle for each limb was identified by examining the kinematic profile of the 1st metatarsal marker, as initial contact was made with the forefoot. Gait events for the subsequent trials were identified using an automatic event identification pipeline command in Visual 3D, and also checked manually.

Temporal-spatial and sagittal plane kinematic data are presented, including frontal plane pelvis and hip data. All kinetic data were normalised to body mass, with joint moments presented as internal moments [21]. GRF data (N/kg), load and decay rate (N/kg/s) are presented relative to the stance phase; sagittal joint moments (Nm/kg) and powers (W/kg) are presented relative to the gait cycle (GC). Throughout this paper, we have defined the following stair ascent sub-phases, according to McFadyen and Winter (1988), and assuming a stance: swing ratio of 60:40: weight acceptance (~0–10% of the GC), pull-up (~10–32% of the GC), forward continuance (~32–60% of the GC), foot clearance (~60–80% of the GC), and foot placement (~80–100% of the GC). The sub-phases were used to label joint power bursts according to McFadyen and Winter (1988). The following joint power bursts occurred during these sub-phases: weight acceptance (H1, K1, A1), pull-up (A2), forward continuance (K2, A3), foot clearance (H3, K3) and foot placement (H4, K4).

## Statistical analysis

The Stata statistical computer package v15.0 (Stata Corp, Texas, USA) was used to carry out normal distribution testing and multivariate regression analyses. Normal distribution of the data was confirmed using skewness to measure the asymmetry and kurtosis to determine the 'peakedness' in histogram of residuals [22, 23].

Two regression models were created to investigate the explained variance in gait biomechanics parameters. Temporal-spatial data and the peak values of joint angles, GRFs, joint moments and powers (averaged from each participant's 10 trials) were treated as dependant variables while comfortable stair ascent speed and femoral neck T-score were considered to be the predictor (independent) variables. Collinearity between T-score and speed was examined by testing the variance inflation factor (VIF). The VIF was determined to be 1.05 which was within the acceptable limit (<10) [24]. The first regression model used stair ascent speed as the predictor variable. The femoral neck T-score was added to the second regression model using

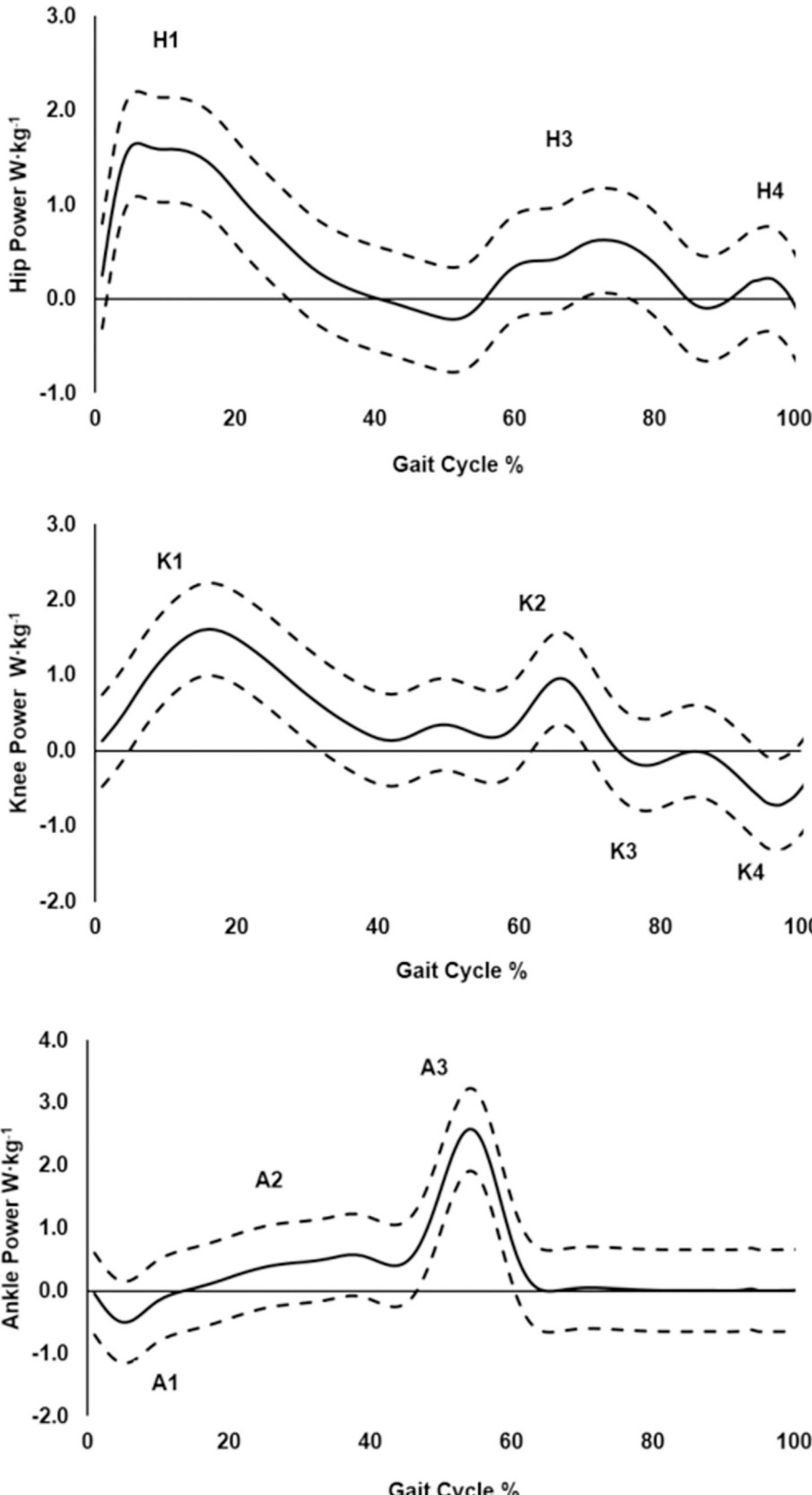

**Fig 2. Ensemble mean (± 1SD: Dashed lines) of lower limb joint powers during stair ascent.** Power bursts are labelled (H1-H4, K1-K4, A1-A3) according to McFadyen and Winter (1988) [18]. Positive [+ve] joint power values indicate power generation while negative [-ve] values indicate power absorption.

blockwise-entry method to explore the relationships between the predictor variables and biomechanics parameters in our cohort of older postmenopausal females. The coefficient of multiple determination ($R^2$) was used to measure at what level the model explained the variability of the response data in relation to the gradient of the regression line ($B$). The result was deemed statistically significant when $P \leq 0.05$.

## Results

### Temporal-spatial and kinematics

The mean (SD) stair ascent speed 0.65 (0.1) m·s$^{-1}$ explained the variance ($R^2$ = 9–47%, $P \leq 0.05$) in most temporal spatial parameters (Table 2). When T-score was included in the regression model, the shared explanatory power of the model only increased for stride width to ($R^2$ = 20%, $P$ = 0.014), with a slope coefficient of $B$ = 0.01 and 95% CI: [0.001 to 0.019] which can be indicative of high precision of this result (Table 2).

Speed explained ($R^2$ = 9%, $P$ = 0.011) and ($R^2$ = 16%, $P$ = 0.007) of the variance for pelvic drop (during foot clearance) and knee flexion (during the weight acceptance), respectively (Table 2). Adding T-score jointly with speed into the regression model significantly increased the shared explained variance for pelvic hike during pull-up and pelvic drop during foot clearance to ($R^2$ = 19%, $P$ = 0.011) and ($R^2$ = 21%, $P$ = 0.017), hip adduction during pull-up to ($R^2$ = 7%, $P$ = 0.054), and knee range of motion (RoM) in the sagittal plane to ($R^2$ = 6%, $P$ = 0.051) (Table 2).

### Ground reaction force and joint moments

Stair ascent speed significantly explained the variance in peak posterior GRF during pull-up ($R^2$ = 28%, $P \leq 0.001$), first vertical GRF peak (Fz1) during pull-up ($R^2$ = 37%, $P \leq 0.001$), and load rate ($R^2$ = 36%, $P \leq 0.001$) (Table 3). Inclusion of T-score in the second regression model increased the shared explained variance for load rate to ($R^2$ = 41%, $P$ = 0.054) (Table 3). Speed significantly explained of the variance in the hip abductor moment (during pull-up) ($R^2$ = 15%, $P$ = 0.009), hip adductor moment (during foot clearance) ($R^2$ = 22%, $P \leq 0.001$) and knee flexor moment (during forward continuance) ($R^2$ = 15%, $P \leq 0.001$) (Table 3). Adding T-score did not have any significant effects on the explanatory power of the regression models for the joint moments (Table 3).

### Joint powers

A substantial amount of the variance in joint powers was explained by STA speed ($P \leq 0.01$) (Table 4). Speed explained ($R^2$ = 18%, $P$ = 0.004) of the variance in H1 (hip extensor power generation during weight acceptance), ($R^2$ = 13%, $P$ = 0.016) in H3 (hip flexor power generation during foot clearance), ($R^2$ = 14%, $P$ = 0.012) in H4 (hip extensor power generation during foot placement), ($R^2$ = 15%, $P$ = 0.012) in K3 (knee extensor power absorption during foot clearance), ($R^2$ = 32%, $P \leq 0.001$) in K4 (knee flexor power absorption during foot placement), ($R^2$ = 38%, $P \leq 0.001$) in A1 (ankle plantarflexor power absorption during weight acceptance) and ($R^2$ = 19%, $P \leq 0.001$) in A2 (ankle plantarflexor power generation during pull-up). Inclusion of T-score in the regression model did not explain any additional variance in joint powers. Ensemble mean (SD) joint power profiles of the participants during stair ascent are presented Fig 2.

**Table 2. Explained variance ($R^2$) and slope coefficient for temporal-spatial and joint kinematics during stair ascent.**

| Gaitparameter | Mean (SD) | Predictor variable | $R^2$% | Predictor variable | Slope coefficient (B) | 95% Confidence interval |
|---|---|---|---|---|---|---|
| Stride width (m) | 0.08 (0.02) | GS | 9 | GS | 0.04* | 0.007: 0.08 |
| | | GS & TS | 20 | TS | 0.01** | 0.001: 0.01 |
| Cycle time (s) | 1.15 (0.15) | GS | 47 | GS | -0.54*** | -0.72: -0.37 |
| | | GS & TS | 47 | TS | -0.008 | -0.04: 0.04 |
| Stance phase (%) | 68 (12) | GS | 46 | GS | -0.44*** | -0.59: -0.29 |
| | | GS & TS | 47 | TS | 0.01 | -0.02: 0.04 |
| Double limb support time (s) | 0.22 (0.07) | GS | 35 | GS | -0.21*** | -0.30: -0.12 |
| | | GS & TS | 35 | TS | -0.006 | -0.02: 0.02 |
| Degrees (°) | | | | | | |
| Pelvic obliquity hike (Pull-up) | 6.84 (1.24) | GS | 5 | GS | 1.52 | -0.42: 3.47 |
| | | GS & TS | 19 | TS | 0.56** | -0.56: 3.10 |
| Pelvic obliquity drop (Foot clearance) | -6.37 (1.51) | GS | 9 | GS | -2.37* | -4.67: -0.07 |
| | | GS & TS | 21 | TS | -0.63** | -4.27: 0.09 |
| Pelvic anterior tilt (Foot clearance) | 11.25 (5) | GS | 5 | GS | -5.82 | -13.73: 2.07 |
| | | GS & TS | 6 | TS | -0.36 | -2.23: 1.51 |
| Hip adduction (Pull-up) | 13.29 (2.6) | GS | 1 | GS | 0.72 | -3.37: 4.81 |
| | | GS & TS | 7 | TS | -0.86* | -1.82: 0.08 |
| Hip abduction (Foot clearance) | -4.77 (2.71) | GS | 1 | GS | -1.16 | -5.46: 3.13 |
| | | GS & TS | 2 | TS | -0.38 | -1.40: 0.64 |
| Hip frontal RoM | 18.06 (3.2) | GS | 1 | GS | 1.66 | -3.52: 6.86 |
| | | GS & TS | 2 | TS | -0.42 | -1.65: 0.80 |
| Hip extension (Forward continuance) | -5.35 (5.15) | GS | 1 | GS | -3.50 | -13.25: 6.24 |
| | | GS & TS | 1 | TS | 0.22 | -2.11: 2.56 |
| Hip flexion (Foot placement) | 71.4 (6.14) | GS | 5 | GS | -7.15 | -16.7: 2.39 |
| | | GS & TS | 5 | TS | -0.66 | -2.94: 1.61 |
| Hip sagittal RoM | 66.08 (4.74) | GS | 2 | GS | -3.68 | -11.15: 3.79 |
| | | GS & TS | 5 | TS | -0.88 | -2.65: 0.88 |
| Knee flexion (Weight acceptance) | 70.03 (5.49) | GS | 16 | GS | -11.29*** | -19.32: -3.27 |
| | | GS & TS | 16 | TS | 0.15 | -1.77: 2.07 |
| Knee flexion (Foot clearance) | 108.8 (6.93) | GS | 1 | GS | -0.68 | -11.73: 10.35 |
| | | GS & TS | 3 | TS | -1.43 | -4.04: 1.18 |
| Knee sagittal RoM | 94.8 (7.08) | GS | 1 | GS | -0.99 | -12.28: 10.28 |
| | | GS & TS | 6 | TS | -2.19* | -4.81: 0.42 |
| Ankle dorsiflexion (Pull-up) | 18.73 (3.6) | GS | 3 | GS | -3.47 | -9.11: 2.15 |
| | | GS & TS | 4 | TS | -0.15 | -1.50: 1.19 |
| Ankle plantarflexion (Foot clearance) | -15.5 (4.78) | GS | 1 | GS | -0.44 | -8.22: 7.33 |
| | | GS & TS | 1 | TS | -0.52 | -2.32: 1.26 |
| Ankle sagittal RoM | 34.28 (4.8) | GS | 2 | GS | -3.16 | -11.59: 5.25 |
| | | GS & TS | 2 | TS | 0.44 | -1.54: 2.43 |

Slope coefficients (B) are presented for stair ascent gait speed (GS) and T-score (TS). Significant findings areas were shaded whereby the point estimate of the regression slope (B) was significantly different from 0 at the following alpha levels

* $P \leq 0.05$

** $P \leq 0.01$, and

*** $P \leq 0.001$

**Table 3. Explained variance ($R^2$) and slope coefficient for peak ground reaction forces and peak joint moment during stair ascent.**

| Gait parameter | Mean (SD) | Predictor variable | $R^2$% | Predictor variable | Slope coefficient (B) | 95% Confidence interval |
|---|---|---|---|---|---|---|
| GRF (N·kg$^{-1}$) & loading/decay rates (N·kg$^{-1}$·s$^{-1}$) | | | | | | |
| Posterior GRF (Pull-up) | -0.03 (0.03) | GS | 28 | GS | -0.15*** | -0.23: -0.07 |
| | | GS & TS | 28 | TS | 0.001 | -0.009: 0.01 |
| Anterior GRF (Forward continuance) | 0.12 (0.01) | GS | 1 | GS | 0.01 | -0.02: 0.06 |
| | | GS & TS | 2 | TS | -0.001 | -0.007: 0.004 |
| 1st vertical GRF peak (Fz1) (Pull-up) | 1.03 (0.08) | GS | 37 | GS | 0.48*** | 0.29: 0.68 |
| | | GS & TS | 41 | TS | -0.01 | -0.04: 0.005 |
| 2nd vertical GRF peak (Fz2) (Forward continuance) | 1.26 (0.11) | GS | 1 | GS | 0.11 | -0.23: 0.47 |
| | | GS & TS | 1 | TS | -0.007 | -0.05: 0.03 |
| Load rate | 5.26 (1.8) | GS | 36 | GS | 10.63*** | 6.25: 15.02 |
| | | GS & TS | 41 | TS | -0.50* | -1.03: 0.03 |
| Decay rate | -10.10 (1.65) | GS | 4 | GS | 3.50 | -1.44: 8.44 |
| | | GS & TS | 6 | TS | -0.26 | -0.88: 0.35 |
| N·m·kg$^{-1}$ | | | | | | |
| Hip abductor (Pull-up) | 0.55 (0.25) | GS | 15 | GS | 0.47** | 0.14: 0.85 |
| | | GS & TS | 18 | TS | -0.04 | -0.13: 0.03 |
| Hip adductor (Foot clearance) | -0.12 (0.05) | GS | 22 | GS | -0.12*** | -0.20: -0.04 |
| | | GS & TS | 23 | TS | -0.006 | -0.02: 0.01 |
| Hip extensor (Weight acceptance) | 0.91 (0.2) | GS | 1 | GS | 0.002 | -0.36: 0.36 |
| | | GS & TS | 1 | TS | 0.31 | -0.05: 0.11 |
| Hip flexor (Forward continuance) | -0.48 (0.2) | GS | 4 | GS | 0.26 | -0.11: 0.64 |
| | | GS & TS | 8 | TS | -0.05 | -0.14: 0.03 |
| Knee extensor (Pull-up) | 0.83 (0.2) | GS | 1 | GS | 0.18 | -0.10: 0.47 |
| | | GS & TS | 5 | TS | -0.02 | -0.09: 0.04 |
| Knee flexor (Forward continuance) | -0.23 (0.1) | GS | 15 | GS | -0.25*** | -0.44: -0.06 |
| | | GS & TS | 20 | TS | 0.03 | -0.01: 0.07 |
| Ankle plantarflexor (Forward continuance) | 1.28 (0.2) | GS | 1 | GS | 0.07 | -0.17: 0.32 |
| | | GS & TS | 1 | TS | 0.01 | -0.04: 0.07 |

Slope coefficients (B) are presented for stair ascent gait speed (GS) and T-score (TS). Significant findings areas were shaded whereby the point estimate of the regression slope (B) was significantly different from 0 at the following alpha levels

\* $P \leq 0.05$

\*\* $P \leq 0.01$, and

\*\*\* $P \leq 0.001$

## Discussion

To our knowledge, this study is the first to explore the relationships between stair ascent speed, T-score and biomechanical gait parameters of older postmenopausal women with BMD levels ranging from healthy to osteoporotic. As hypothesised, speed remained the most

**Table 4. Explained variance ($R^2$) and slope coefficient for peak joint powers during stair ascent.**

| Joint power parameter | Mean (SD) | Predictor variable | $R^2$% | Predictor variable | Slope coefficient (B) | 95% Confidence interval |
|---|---|---|---|---|---|---|
| | | | W·kg$^{-1}$ | | | |
| H1 (Weight acceptance) | 2.18 (0.88) | GS | 18 | GS | 1.94*** | 0.66: 3.22 |
| | | GS & TS | 18 | TS | -0.67 | -0.37: 0.23 |
| H3 (Foot clearance) | 0.79 (0.28) | GS | 13 | GS | 0.51** | 0.10: 0.92 |
| | | GS & TS | 13 | TS | 0.02 | -0.07: 0.12 |
| H4 (Foot placement) | 0.45 (0.36) | GS | 14 | GS | 0.69** | 0.16: 1.23 |
| | | GS & TS | 14 | TS | -0.01 | -0.14: 0.11 |
| K1 (Weight acceptance) | 1.81 (0.56) | GS | 4 | GS | 0.59 | -0.27: 1.46 |
| | | GS & TS | 7 | TS | -0.11 | -0.32: 0.08 |
| K2 (Forward continuance) | 1.3 (0.69) | GS | 1 | GS | 0.38 | -0.71: 1.47 |
| | | GS & TS | 2 | TS | 0.08 | -0.18: 0.34 |
| K3 (Foot clearance) | -0.4 (0.23) | GS | 15 | GS | -0.44** | -0.77: -0.11 |
| | | GS & TS | 15 | TS | 0.18 | -0.06: 0.09 |
| K4 (Foot placement) | -0.76 (0.5) | GS | 32 | GS | -1.48*** | -2.16: -0.81 |
| | | GS & TS | 34 | TS | 0.06 | -0.15: 0.46 |
| A1 (Weight acceptance) | -0.68 (0.5) | GS | 38 | GS | -1.71*** | -2.38: -1.04 |
| | | GS & TS | 39 | TS | 0.07 | -0.08: 0.23 |
| A2 (Pull-up) | 0.99 (0.19) | GS | 19 | GS | 0.75*** | 0.27: 1.23 |
| | | GS & TS | 19 | TS | -0.01 | -0.12: 0.10 |
| A3 (Forward continuance) | 3.19 (1.05) | GS | 1 | GS | 0.60 | -1.05: 2.26 |
| | | GS & TS | 1 | TS | 0.06 | -0.33: 0.46 |

The joint power bursts are labelled (H1-H4, K1-K4, A1-A3) according to McFadyen and Winter (1988) [18].

Slope coefficients (B) are presented for stair ascent gait speed (GS) and T-score (TS). Significant findings areas were shaded whereby the point estimate of the regression slope (B) was significantly different from 0 at the following alpha levels

* $P \leq 0.05$

** $P \leq 0.01$, and

*** $P \leq 0.001$

important variable to explain most of the variance for temporal-spatial, GRFs, joint kinematic and kinetic data. When T-score was introduced into the regression model, it explained the variance in some gait parameters associated with dynamic stability in both the sagittal and frontal planes.

A review of the literature has found that only a few studies reported speed or cadence during stair ascent [25–34], making it difficult to compare the findings from different studies. This is because our findings have demonstrated that self-selected stair ascent speed is an influential predictor variable. In the present study, participants' mean (SD) stair ascent speed was 0.65 (0.10) m·s$^{-1}$. Another gait study involving older women (mean age = 82.2 years) who climbed a 5-step staircase reported an average speed of 0.49 (0.13) m·s$^{-1}$ [25], while a different study investigating younger women (mean age = 23.9 years) ascending a 3-step staircase reported a speed of 0.65 (0.03) m·s$^{-1}$ [26]. These findings indicate that our participants were functioning at a high physical level despite their age and varied bone health. Our participants also exhibited a high cadence of 107 (17) steps/min, compared to another study [27] involving older women (mean age = 74.9 years) climbing a 4-step staircase at a rate of 92 (10) steps/min. The fast stair ascent speed in our study was attributed to the high cadence, and not step length, as step length was naturally constrained by the staircase dimensions.

The first regression model with speed as the only predictor variable significantly explained the variance in many of the temporal-spatial parameters (e.g., stride width, cycle time, stance phase and double limb support time) and only two of the joint angles (e.g., pelvic drop and knee flexion). Similar to our previous research, where we explored the relationship between speed, T-score and gait parameters during level walking [11], gait speed was found to be one of the most important predictor variables when analysing the gait biomechanics in postmenopausal women with varied BMD. Therefore, we recommend that future studies quantify stair ascent speed, and specify whether it is self-selected/comfortable or standardised, when examining the effects of other predictor variables, such as BMD, on gait parameters.

Older adults have demonstrated a tendency to increase their stride width to enhance dynamic stability during level walking [11, 35]. In line with our previous study on level walking [11], we observed increased stride width during stair ascent in the same group of postmenopausal women (Table 2). The T-score explained additional variance in stride width (Table 2) associated with low BMD or osteoporosis. Considering the biomechanical challenges and falls risk associated with stair climbing, it is not surprising to observe an increased base of support amongst participants with low bone mineral density. It should be acknowledged that participants were aware of their T-score and bone health, so it was possible that increased stride width served as a compensatory strategy to improve dynamic stability (and therefore reduce falls risk) during stair ascent. We are unable to establish the cause and effect relationship due to the cross-sectional nature of study design. However, we believe load-bearing exercises that challenge stride width during dynamic tasks (e.g., balancing with a narrow base of support on a compliant or moving surface) should be incorporated into an exercise programmes tailored to people with low BMD.

Combining the T-score with speed in the second regression model increased the explanatory power of models for pelvic hike and drop, and hip adduction during pull-up (Table 2). These parameters are related to pelvic control and dynamic stability in the frontal plane during stair ascent, and demand sufficient strength from the hip abductors and adductors and the deeper core muscles of the trunk. These findings suggest older women with low BMD may benefit from weight-bearing exercise programmes aimed at strengthening these muscle groups to enhance dynamic stability during more challenging ambulatory tasks. The complex interaction of the hip abductor and adductors, which have multiple points of insertion on the femur, may generate a strain distribution responsible for driving bone remodelling. These muscles, primarily the gluteus medius and minimus, may also help to compress the entire femoral neck due to co-contraction of the surrounding musculature. The compressive stress due to this co-contraction may be structurally beneficial to induce bone remodelling in the femur [36] and help maintain bone density in this population.

Significant slope coefficient for the first regression model (speed as the only predictor) showed a linear and positive relationship between stair ascent speed with the first vertical GRF peak (Fz1) and load rate (Table 3). Inclusion of T-score increased the explanatory power of the regression model for load rate, indicating older women with low BMD tended to load the lower limbs less during the stair ascent task. This may be a compensatory strategy to decrease the effect of cyclic loading during stair walking, which is a more challenging task than level walking, with a higher risk for bone micro-fractures. It is well understood that bone remodelling is stimulated by the loading rate [37, 38] and physical activities with loading intensity beyond certain level of threshold (10–15 BW/s) improve bone density in healthy women [39]. Although high-impact exercises have been reported to improve bone density in older females with low BMD [40], it should be acknowledged that loading at high rates may increase the risk of bone micro-fracture in older postmenopausal women and should not be performed if dynamic stability is challenged and the risk of falls is increased.

Addition of T-score to the regression model did not increase the explanatory power of the model for the lower limb joint power bursts. However, stair ascent speed significantly explained the variance in hip and ankle peak joint power generation (H1, H3, H4, A2) (Table 4) and in knee and ankle peak joint power absorption (K3, K4, A1) (Table 4). To our knowledge, only two studies have previously explored the three-dimensional biomechanics of postmenopausal women with low BMD during level walking [11, 12]. One of these studies [12] identified H1, H2, H3, K4 and A2 as the main gait parameters to predict low BMD. However, it is not possible to compare their results with the current study because the two studies explored different locomotor tasks and one study [12] did not report or include gait speed in their statistical models. The current results indicate that older postmenopausal women are capable of producing sufficient joint power outputs regardless of their level of BMD. Walking at a faster pace (beyond comfortable walking speed) during uphill activities, including stair ascent, could provide the mechanical stimulus to positively influence the stress/strain distribution on the long bones of the lower limbs. However, care should be taken during stair negotiation and light handrail use could help with safety without compromising mechanical stimuli and stability.

Some limitations of this study must be acknowledged. Based on the WHO definition [17], 81% of the participants were considered to be active, as many of them were engaged in regular physical activity (Table 1). It is possible that a group of older postmenopausal women, leading a more sedentary lifestyle, could exhibit slower stair ascent speed and different gait outcomes. Our findings may not be generalisable to men and younger women with low BMD. Finally, due to the cross-sectional nature of this study, cause and effect of gait biomechanics in relation to BMD (T-score) could not be identified.

## Conclusion

Most of the gait variance during stair ascent was explained by speed, while T-score mainly explained the variance in gait parameters related to dynamic stability associated with hip and pelvic kinematics in the frontal plane. Furthermore, our findings suggest increased stride width and decreased load rate could be associated with low BMD. Based on these findings, a weight-bearing exercise programme that incorporates a combination of balance activities and hip abductor/adductor strengthening could improve dynamic stability to benefit older women with low BMD during more challenging daily tasks, such as stair walking.

## Supporting information

**S1 Data.**
(XLSX)

## Author Contributions

**Conceptualization:** Ali Dostan, Catherine A. Dobson, Natalie Vanicek.

**Formal analysis:** Ali Dostan.

**Funding acquisition:** Catherine A. Dobson.

**Investigation:** Ali Dostan.

**Methodology:** Ali Dostan.

**Project administration:** Ali Dostan.

**Supervision:** Catherine A. Dobson, Natalie Vanicek.

**Writing – original draft:** Ali Dostan, Catherine A. Dobson, Natalie Vanicek.

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
