## [Decision Letter · Decision Letter 0]

7 Sep 2022

PONE-D-22-11086Stair ascent in postmenopausal women: A regression analysis exploring the relationships between walking speed and T-score with sagittal and frontal plane gait biomechanicsPLOS ONE

Dear Dr. Dostan,

Thank you for submitting your manuscript to PLOS ONE. After careful consideration, we feel that it has merit but does not fully meet PLOS ONE’s publication criteria as it currently stands. Therefore, we invite you to submit a revised version of the manuscript that addresses the points raised during the review process.

The reviewers have raised a number of questions related to the study design and statistical approach that need to be adequately addressed.

We look forward to receiving your revised manuscript.

Kind regards,

John Leicester Williams, Ph.D.

Academic Editor

PLOS ONE

“We would like to thank and acknowledge Osteoporosis Research in East Yorkshire (OSPREY) (charity commission number: 1013289) who funded this research.”

“AD, CAD, NV received funding from Osteoporosis Research in East Yorkshire (OSPREY) (charity commission number: 1013289).

There are no grant number.

URL:https://www.osprey.org.uk/index.php

Reviewers' comments:

Reviewer's Responses to Questions

**Comments to the Author**

1. Is the manuscript technically sound, and do the data support the conclusions?

Reviewer #1: Partly

Reviewer #2: Yes

2. Has the statistical analysis been performed appropriately and rigorously? 

Reviewer #1: No

Reviewer #2: Yes

3. Have the authors made all data underlying the findings in their manuscript fully available?

Reviewer #1: Yes

Reviewer #2: Yes

4. Is the manuscript presented in an intelligible fashion and written in standard English?

Reviewer #1: Yes

Reviewer #2: Yes

5. Review Comments to the Author

Reviewer #1: The manuscript examined the relationship between the gait related biomechanical variables and stair ascent speed and T-scores using multiple regression analyses. The study included relatively large number of participants. However, I have major reservations for the research question and analysis method of the regression model and variable selection method.

General comments –

1. The main hypothesis does not match the purpose of the student. It is well known that gait speed is highly correlated with many biomechanical gait variables. It is not worthwhile to investigate the relationship between the gait speed and gait related variables, even though it has not been explored extensively in the population as stated by the authors.

2. The regression analysis used in the study is sufficient enough to answer the research question. The inclusion of all variables (Tables 2-4) in the analyses are not appropriate without proper justifications as many of the included variables are more likely highly correlated with each other. Such an approach is not that different from running correlation coefficients among the variables. Correlation analysis is not sufficient in addressing the research question. More importantly, inclusion of all lower limb biomechanical variables and special and temporal gait variables in the analyses is too broad without focus. There are many stair gait biomechanical research studies and the authors should be able to draw sufficient information to narrow down the more important variables related to hip biomechanics that are more relevant to the hip fracture related to osteoporosis.

3. The gait speed determination was not provided in the method section but it was used as a main predictor of the regression models.

4. It seems that the authors included data from both limbs, trailing limb on the second step and leading limb in the third step, in their analysis. Did the authors used 10 individual trials and both limb’s data in the regression analysrs? This may inflate the sample size and violation of regression analysis as individual trials of the same condition for an individual participant are not considered independent of each other.

Specific comments –

Ln 74-76 The hypothesis is not a novel one as it is commonly known that gait speed is a main factor for changes in gait biomechanics, especially the gait kinetics.

Ln 84-85 the sample size is relatively small for the osteoporosis patients.

Ln 100-101 A picture of the stair system would be helpful.

Ln 105 It is important to have a consistent stair ascent speed. Did the authors monitor the speed?

Ln 105 Did the authors used the data of the average of 10 trials or data of the 10 dividual trials in the regression analyses? See also my general comment on this.

Ln 114 Change this to “Inverse dynamics analysis”.

Ln 118-121 It is not clear how the phases of stair ascent was defined. Later the authors referred to “pull-up” and “load rate” which were never mentioned that how there phases were defined. Although they referred to McFadyen and Winter (1988), but these should be defined in the methods.

Reviewer #2: General comments

The submitted manuscript is an observational study aimed at investigating the gait biomechanics of stair ascent for older, postmenopausal women. Forty-five women underwent DEXA to assessment bone health quality, and were subsequently categorised as either: healthy, osteopenic, or osteoporotic. On a separate occasion they then ascended aa five-step stair to capture kinematics and kinetics. Finally, regression analyses were conducted, revealing that gait speed was the most important characteristic in explaining variance in all biomechanical measures. T-score, or bone density score, explained variance for gait measures relating to dynamic stability, and has implications targeting the hip adductors and abductors, and core musculature for future exercise interventions for falls prevention in older women.

The study methodology is appropriate to satisfy the aims, the findings are original and would interest the journal’s readership.

Specific comments

Title

The title is currently verbose, and particularly throughout the manuscript, the use of ‘T-score’ may lose the reader. Although correct and scientific, many PLOS One readers from the natural and social sciences may not understand what T-score is without explanation.

Introduction

The aim could be written with greater clarity, and appears not to directly transfer to the aim stated in the abstract.

Methods

Participants

Page 4

I recommend presenting data for the measurement error associated with the DEXA scans. A minor point would also be to add the % proportion of ‘healthy’, ‘osteopenic’, etc.

You mention ‘strict criteria’ were used to exclude potential participants. However, the specific inclusion and exclusion criteria is not clearly stated, nor is there any information on whether i) any did not complete testing, ii) there were any adverse events (particularly trips), or iii) data was lost incomplete/lost for any participants. For the latter, if so, then what methods were used for data handling.

Protocol

Page 5

Consider including a pictorial / reproduction diagram of the stair set-up, including the gait cycle. This would benefit the paper in readability and help interpret the precise gait cycle.

Data analysis

Page 5

There is scant information on how joint moments and powers were calculated. Please expand on the inverse dynamic analyses. Also, the specific gait cycle could be better described, specifying the left / right foot contact for each gait phase.

Page 6

You give reference to McFadyen and Winter (1988), but there is no explanation of STA, H1, H2, etc, in the Method. These need succinctly adding.

Results

The results for joint power are not written in an accessible, easily interpretable manner. The use of H1, etc, do not really illustrate the gait cycle events. This subsection could be clearer.

Also, include exact P values and check that the significant variables highlighted in tables 2 and 3 are consistent with descriptions in the text.

Discussion

Page 12

I recommend briefly confirming whether limb length was an influential factor in step cadence and other temporo-spatial characteristics.

Page 113, line 222: “we recommend future studies quantify stair ascent speed…”, and based on the preceding sentence walking gait too.

Line 226: typo “level walk”

Line 241-244: It would be insightful to illustrate the trunk’s role in the ascent gait cycle (e.g. stance) for interpretation.

Page 15

The authors highlight the importance of physical activity in human gait in their limitations, but given there is a wealth of evidence to support those ‘highly’ physical active or ‘active’ in having greater gait ability, it is worth providing data on how many (n, %) were classified as: very active, active, low active, or sedentary.

6. PLOS authors have the option to publish the peer review history of their article (what does this mean?). If published, this will include your full peer review and any attached files.

Reviewer #1: No

Reviewer #2: **Yes: **James P. Gavin

---

## [Author Response · Author response to Decision Letter 0]

30 Dec 2022

Response to the editor's specific comments: 

We would like to confirm that our revised manuscript meets the PLOS ONE's style requirements. 

There are no grant numbers associated with the funding that we have received as this research was funded through a charitable donation by Osteoporosis Research in East Yorkshire (OSPREY) (charity commission number: 1013289). 

Could you please amend the funding statement to: “This research was funded through a charitable donation by Osteoporosis Research in East Yorkshire (OSPREY) (charity commission number: 1013289). The funders had no role in study design, data collection and analysis, decision to publish, or preparation of the manuscript.”

We have also amended the acknowledgment section to read “We would like to thank Osteoporosis Research in East Yorkshire (OSPREY) (charity commission number: 1013289) for their support.” 

We would also like to confirm that we have now uploaded the minimal anonymized data set that is required to replicate the results of this study. 

Response to the reviewer's specific comments:

Specific Comments Reviewer #1: 

Ln 74-76 - The hypothesis is not a novel one as it is commonly known that gait speed is a main factor for changes in gait biomechanics, especially the gait kinetics.

The focus of the study is not to study the effects of gait speed on the biomechanics of this population. Instead we focused on identifying the level at which bone mineral density (T-score) explained the variance in gait parameters by also considering the effects of gait speed. 

We used gait speed as one of our independent variables to consider its effects when investigating the relationship between gait parameters and T-score (level of bone density). Without the inclusion of gait speed in our regression model and consideration for its effect, we may run the risk of inflated results similar to a previous study published by ElDeeb and Khodair (2014) (ElDeeb & Khodair, 2014).

ElDeeb and Khodair (2014) reported many gait parameters such as hip adductor and extensor moments, and joint power bursts (H1, H2, H3, K4 and A2) to be significant predictors for low bone density in postmenopausal women without accounting for the effects of walking speed. In addition to that, our previous study (2) demonstrated that gait speed was an important predictor variable and must be taken into account when considering other factors that could affect gait performance (e.g. bone density (T-score), age, etc). 

Ln 84-85 the sample size is relatively small for the osteoporosis patients.

Our participants included 26 women with osteopenia and 6 with osteoporosis. Therefore 71% of our participants have low BMD scores (We have included a statement, please see P4, lines 90-92). The low number of participants with osteoporosis does not affect our statistics/results as the participants were not assigned into different groups. Instead, we conducted a multi-regression analysis and studied the level of explained variance for each gait parameter. We then identified gait parameters that were related to participants with low BMD when T-score significantly explained the variance in the gait parameter. 

Ln 100-101 A picture of the stair system would be helpful.

Additional figure has been added (figure 1). 

Ln 105 It is important to have a consistent stair ascent speed. Did the authors monitor the speed?

Yes, the participants were instructed to climb the stairs at their comfortable preferred walking speed to capture data related to their usual biomechanics and to avoid any potential falls related to walking too quickly or slowly.

Ln 105 Did the authors used the data of the average of 10 trials or data of the 10 dividual trials in the regression analyses? See also my general comment on this.

The average value for the trials was used. 

Ln 114 Change this to “Inverse dynamics analysis”.

This has been amended (please see P6, line 126)

Ln 118-121 It is not clear how the phases of stair ascent was defined. Later the authors referred to “pull-up” and “load rate” which were never mentioned that how there phases were defined. Although they referred to McFadyen and Winter (1988), but these should be defined in the methods.

We have now introduced the terms in the method section. (Please see 7 lines 147- 154)

Specific Comments Reviewer #2: 

Title

The title is currently verbose, and particularly throughout the manuscript, the use of ‘T-score’ may lose the reader. Although correct and scientific, many PLOS One readers from the natural and social sciences may not understand what T-score is without explanation.

The title has been amended. 

We have included a description to define T-score in the method section in P4, lines 86-88.

Introduction

The aim could be written with greater clarity, and appears not to directly transfer to the aim stated in the abstract.

This has been amended (please see P3, lines 74-75)

Methods

Participants

Page 4

I recommend presenting data for the measurement error associated with the DEXA scans. A minor point would also be to add the % proportion of ‘healthy’, ‘osteopenic’, etc.

You mention ‘strict criteria’ were used to exclude potential participants. However, the specific inclusion and exclusion criteria is not clearly stated, nor is there any information on whether i) any did not complete testing, ii) there were any adverse events (particularly trips), or iii) data was lost incomplete/lost for any participants. For the latter, if so, then what methods were used for data handling.

The reliability of DEXA measurements for bone mineral density are reported to be (r = 0.98) during supine scanning (Lohman et al., 2009). We agree that due to the differences in X-ray energy generation/absorption and bone edge detection paradigms, the BMD level reported in g/cm2 differs amongst DEXA manufacturers. To avoid these issues, we used the same DEXA scanning device throughout the study and utilised the T-score to report BMD level which provides a normalised value.

We have added a sentence to state the number of healthy and low BMD participants in percentage (please see P4, lines 91-92). 

The exclusion criteria are listed P4, lines 93-96 as the following: “gait abnormalities, neurological disorders, any cardiac failure, or if they had received a treatment course of hormone replacement therapy, glucocorticoids, teriparatide and/or bisphosphonate within the five years prior to the study enrolment”. 

We have now added further information in the “participants” section (P4, lines 84-86) to describe our inclusion criteria. “Our inclusion criteria included postmenopausal females aged between 65-70 years with a BMI between 18-30 kg/m2 and various levels of BMD (ranging from +1 to −3 T-score).” 

i) We can confirm that every participant successfully completed the study during a single visit to our laboratory. Please see P5 lines 105-106.

ii) All participants were otherwise healthy and there were no falls or any other incident to be reported. Please see P5 line 117.

iii) Marker dropout, ghost markers, and marker movement were the more prominent notes. To fix the issue marker trajectory data were interpolated using a cubic-spline algorithm. This was referred to in “data analysis” section, P6, line 121. 

Protocol

Page 5

Consider including a pictorial / reproduction diagram of the stair set-up, including the gait cycle. This would benefit the paper in readability and help interpret the precise gait cycle.

A figure has now been included (figure 1). 

Data analysis

Page 5

There is scant information on how joint moments and powers were calculated. Please expand on the inverse dynamic analyses. Also, the specific gait cycle could be better described, specifying the left / right foot contact for each gait phase.

We have added a description on P6, lines 126- 131. 

We have now included more information in P6, line 137-141. Please also see figure 1 for further information. 

Page 6

You give reference to McFadyen and Winter (1988), but there is no explanation of STA, H1, H2, etc, in the Method. These need succinctly adding.

We have now introduced the terms in the method section. (Please see P7 lines 147-154)

Results

The results for joint power are not written in an accessible, easily interpretable manner. The use of H1, etc, do not really illustrate the gait cycle events. This subsection could be clearer.

Also, include exact P values and check that the significant variables highlighted in tables 2 and 3 are consistent with descriptions in the text.

We have described the joint power bursts with the corresponding gait cycle event for these terms, e.g. H1 (hip extensor power generation during weight acceptance). 

We have now reported the exact P-vales and double checked the descriptions in Table 2 and 3 are consistent with text.

Discussion

Page 12

I recommend briefly confirming whether limb length was an influential factor in step cadence and other temporo-spatial characteristics.

The data are normalised to the participant’s height therefore limb length should not affect cadence or any other temporal-spatial data. 

Page 113, line 222: “we recommend future studies quantify stair ascent speed…”, and based on the preceding sentence walking gait too.

That’s correct, however, we would like to keep the focus of this study on stair climbing. We have previously published a paper with a focus on level walking gait (Dostanpor et al., 2018) where we discussed the effects of walking speed and low BMD on gait parameters.

Line 226: typo “level walk”

Amended. 

Line 241-244: It would be insightful to illustrate the trunk’s role in the ascent gait cycle (e.g. stance) for interpretation.

Unfortunately, we only focused on the lower limb and we do not have any information about the trunk. We now collect trunk and upper limb motion regularly to ensure sufficient data from trunk and upper limbs are collected for our current and future studies.

Page 15

The authors highlight the importance of physical activity in human gait in their limitations, but given there is a wealth of evidence to support those ‘highly’ physical active or ‘active’ in having greater gait ability, it is worth providing data on how many (n, %) were classified as: very active, active, low active, or sedentary.

We have used the WHO guidelines on physical activity and sedentary behaviour to report the (self-reported) activity levels of our participants. Please see P4, lines 99-101, and P15 lines 310-311.

References: 

Benedetti, M. G., Furlini, G., Zati, A., & Mauro, G. L. (2018). The Effectiveness of Physical Exercise on Bone Density in Osteoporotic Patients. In BioMed Research International. https://doi.org/10.1155/2018/4840531

Dostanpor, A., Dobson, C. A., & Vanicek, N. (2018). Relationships between walking speed, T-score and age with gait parameters in older post-menopausal women with low bone mineral density. Gait and Posture, 64, 230–237. https://doi.org/10.1016/j.gaitpost.2018.05.005

ElDeeb, A. M., & Khodair, A. S. (2014). Three-dimensional analysis of gait in postmenopausal women with low bone mineral density. Neuroengineering and Rehabilitation, 11(1), 55. https://doi.org/10.1186/1743-0003-11-55

Frost, H. M. (1994). Wolff’s Law and bone’s structural adaptations to mechanical usage: an overview for clinicians. In Angle Orthodontist (Vol. 64, Issue 3, pp. 175–188). https://doi.org/10.1043/0003-3219(1994)064<0175:WLABSA>2.0.CO;2

Frost, H M. (1990). Skeletal structural adaptations to mechanical usage (SATMU): 1. Redefining Wolff’s law: the bone modeling problem. The Anatomical Record, 226(4), 403–413. https://doi.org/10.1002/ar.1092260402

Frost, Harold M. (2004). A 2003 Update of Bone Physiology and Wolff ’ s Law for Clinicians. 74(1), 3–15.

Hair, J. F., Black, W. C., Babin, B. J., & Anderson, R. E. (1995). Multivariate Data Analysis (3rd ed). New York: Macmillan.

Lohman, M., Tallroth, K., Kettunen, J. A., & Marttinen, M. T. (2009). Reproducibility of dual-energy x-ray absorptiometry total and regional body composition measurements using different scanning positions and definitions of regions. Metabolism: Clinical and Experimental, 58(11). https://doi.org/10.1016/j.metabol.2009.05.023

---

## [Decision Letter · Decision Letter 1]

7 Mar 2023

Relationship between stair ascent gait speed, bone density and gait characteristics of postmenopausal women

PONE-D-22-11086R1

Dear Dr. Dostan,

We’re pleased to inform you that your manuscript has been judged scientifically suitable for publication and will be formally accepted for publication once it meets all outstanding technical requirements.

Kind regards,

John Leicester Williams, Ph.D.

Academic Editor

PLOS ONE

Additional Editor Comments (optional):

Reviewers' comments:

Reviewer's Responses to Questions

**Comments to the Author**

1. If the authors have adequately addressed your comments raised in a previous round of review and you feel that this manuscript is now acceptable for publication, you may indicate that here to bypass the “Comments to the Author” section, enter your conflict of interest statement in the “Confidential to Editor” section, and submit your "Accept" recommendation.

Reviewer #2: All comments have been addressed

2. Is the manuscript technically sound, and do the data support the conclusions?

Reviewer #2: Yes

3. Has the statistical analysis been performed appropriately and rigorously? 

Reviewer #2: Yes

4. Have the authors made all data underlying the findings in their manuscript fully available?

Reviewer #2: Yes

5. Is the manuscript presented in an intelligible fashion and written in standard English?

Reviewer #2: Yes

6. Review Comments to the Author

Reviewer #2: I commend the authors in their clear and logical responses to the reviewers. I feel they have adequately addressed the major and minor comments of the two reviewers.

7. PLOS authors have the option to publish the peer review history of their article (what does this mean?). If published, this will include your full peer review and any attached files.

Reviewer #2: **Yes: **James P. Gavin

---

## [Editor Report · Acceptance letter]

13 Mar 2023

PONE-D-22-11086R1 

Relationship between stair ascent gait speed, bone density and gait characteristics of postmenopausal women 

Dear Dr. Dostan:

I'm pleased to inform you that your manuscript has been deemed suitable for publication in PLOS ONE. Congratulations! Your manuscript is now with our production department. 

Kind regards, 

on behalf of

Dr. John Leicester Williams 

Academic Editor

PLOS ONE